# Relationship of micro-RNA, mRNA and eIF Expression in Tamoxifen-Adapted MCF-7 Breast Cancer Cells: Impact of miR-1972 on Gene Expression, Proliferation and Migration

**DOI:** 10.3390/biom12070916

**Published:** 2022-06-29

**Authors:** Akhil Behringer, Darko Stoimenovski, Martin Porsch, Katrin Hoffmann, Gerhard Behre, Ivo Grosse, Thomas Kalinski, Johannes Haybaeck, Norbert Nass

**Affiliations:** 1Department of Pathology, Medical Faculty, Otto-von-Guericke University Magdeburg, Leipziger Str. 44, D-39120 Magdeburg, Germany; akhil.behringer@st.ovgu.de (A.B.); darko.stoimenovski@st.ovgu.de (D.S.); thomas.kalinski@med.ovgu.de (T.K.); 2Institute of Computer Science, Martin Luther University Halle-Wittenberg, Von-Seckendorff-Platz 1, D-06120 Halle, Germany; martin.porsch@informatik.uni-halle.de (M.P.); ivo.grosse@informatik.uni-halle.de (I.G.); 3Institute of Human Genetics and Medical Biology, Martin Luther University Halle-Wittenberg, Magdeburger Str. 2, D-06112 Halle, Germany; katrin.hoffmann@uk-halle.de; 4Dessau Medical Center and Brandenburg Medical School Theodor Fontane (MHB), Department for Internal Medicine I, Auenweg 38, D-06847 Dessau, Germany; gerhard.behre@klinikum-dessau.de; 5German Centre for Integrative Biodiversity Research (iDiv) Halle-Jena-Leipzig, Von-Seckendorff-Platz 1, D-06120 Halle, Germany; 6Diagnostic & Research Center for Molecular BioMedicine, Institute of Pathology, Medical University Graz, Neue Stiftingtalstrasse 6, A-8010 Graz, Austria; johannes.haybaeck@i-med.ac.at; 7Institute of Pathology, Neuropathology and Molecular Pathology, Medical University of Innsbruck, Müllerstrasse 44, A-6020 Innsbruck, Austria

**Keywords:** breast cancer, tamoxifen, MCF-7, gene expression, eukaryotic initiation factors

## Abstract

Background: Tamoxifen-adapted MCF-7-Tam cells represent an in-vitro model for acquired tamoxifen resistance, which is still a problem in clinics. We here investigated the correlation of microRNA-, mRNA- and eukaryotic initiation factors (eIFs) expression in this model. Methods: MicroRNA- and gene expression were analyzed by nCounter and qRT-PCR technology; eIFs by Western blotting. Protein translation mode was determined using a reporter gene assay. Cells were transfected with a miR-1972-mimic. Results: miR-181b-5p,-3p and miR-455-5p were up-, miR-375, and miR-1972 down-regulated and are significant in survival analysis. About 5% of the predicted target genes were significantly altered. Pathway enrichment analysis suggested a contribution of the FoxO1 pathway. The ratio of polio-IRES driven to cap-dependent protein translation shifted towards cap-dependent initiation. Protein expression of eIF2A, -4G, -4H and -6 decreased, whereas eIF3H was higher in MCF-7-Tam. Significant correlations between tamoxifen-regulated miRNAs and eIFs were found in representative breast cancer cell lines. Transfection with a miR-1972-mimic reverses tamoxifen-induced expression for a subset of genes and increased proliferation in MCF-7, but reduced proliferation in MCF-7-Tam, especially in the presence of 4OH-tamoxifen. Migration was inhibited in MCF-7-Tam cells. Translation mode remained unaffected. Conclusions: miR-1972 contributes to the orchestration of gene-expression and physiological consequences of tamoxifen adaption.

## 1. Introduction

Breast cancer still represents the most common neoplasia in women worldwide, showing increasing incidence. Although this disease has an overall good prognosis, certain subtypes are still challenging and there is a need for the identification of additional therapy target molecules, as well as predictive biomarkers [1].

Breast cancer is clinically classified according to receptor status, proliferation rate and classical pathological parameters, such as size, grading and lymph-node metastasis [2]. Based on gene expression studies, breast cancer can be classified into five molecular intrinsic subtypes, i.e., luminal-A and -B, HER2-overexpressing, the normal-like and the basal-like subtype [3]. These subtypes overlap widely with the clinical classification based on the immuno-histochemical analysis of estrogen- (ER), progesterone-receptor (PR), the epidermal growth factor receptor 2 (HER2/NEU), as well as the proliferation marker ki-67 [4]. The molecular basal-like subtype overlaps widely with the triple-negative subtype defined by the absence of hormone receptors by immunohistochemistry, but can be further divided in 6 subtypes [5]. These types are designated as basal-like 1 (BL1), and -2 (BL2), immunomodulatory (IM), luminal androgen receptor (LAR), mesenchymal stem cell-like (MSL) and mesenchymal (M) [6].

ER-positive tumors are treated by targeting this receptor either directly, using selective estrogen receptor mediators (SERMs), such as tamoxifen or by blocking estrogen biosynthesis with aromatase inhibitors. The use of tamoxifen seems favorable in premenopausal women [7,8]; however, a relapse occurs in about 30% of the tamoxifen-treated patients [9]. This is attributed to several reasons, such as tumor heterogeneity, a mutated estrogen receptor [10] or the so-called acquired tamoxifen resistance [11]. The latter is characterized by a shift from estrogen-driven proliferation to other growth factor pathways, such as epidermal-growth factor (EGF) or insulin-like growth factor-1 (IGF-1) [12]. In addition, estrogen-receptor splice variants, such as the ERα-36 or -42, are discussed as the basis for tamoxifen resistance [13].

Because of this significant number of relapses, there is an ongoing quest for tamoxifen-resistance predicting biomarkers. Having such a biomarker in hand, clinicians could apply alternative therapies early, thereby avoiding relapses under tamoxifen. Such therapies could imply inhibitors of cyclin dependent kinases (CDK) [14] or novel targets may be based on novel predictive biomarkers. A new development in this context is the design of bifunctional degrader molecules for bromodomain-containing proteins, such as BRD3 [15,16], which target the estrogen signaling pathway.

Acquired tamoxifen resistance has repeatedly been investigated using cell culture models [12]. For this approach, the luminal-A breast cancer cell line MCF-7 is adapted to the presence of tamoxifen for at least 12 weeks. As there is considerable variation in the experimental parameters, such as the use of tamoxifen, or its active metabolite 4-OH-tamoxifen, the concentration of these substances and the presence of serum, the outcome showed considerable variation. Nevertheless, most genes regulated in this experimental setting also have significant impact on the prognosis of breast cancer patients. Besides the changes in mRNA expression, alterations in micro RNA content have also been observed. A recent review listed about 75 micro RNAs associated with tamoxifen resistance [17]. For example, mir-375 [18], mir-519a [19] miR-181b [20], miR-363, [21] or miR-451 [22] were functionally associated with tamoxifen resistance.

An important characteristic of cancer cells is the requirement for sufficient protein biosynthesis to allow high proliferation rates. The eukaryotic initiation factors (eIFs) are key control elements for this process [23]. These proteins do not only modulate protein biosynthesis quantitatively but also modify the translation efficiency for certain stress- and cancer related transcripts [24,25]. Accordingly, an altered ratio of cap- versus IRES (internal ribosome entry site)-mediated initiation depending on eIF expression was shown [26,27]. The abundance and function of eIFs have, therefore, been investigated in several cancer entities and breast cancer is no exception [28]. Several reports indicate that miRNAs confer drug-resistance via modulating eIF expression [29,30,31]. We, therefore, hypothesized that miRNA expression changes could be the basis for differential eIF expression in tamoxifen-adapted cells and in acquired tamoxifen resistance.

We have recently reported on expression changes in mRNA and non-coding RNA in our MCF-7 based tamoxifen model [32]. Now, we extend this analysis to micro RNAs (miRNAs) and eIFs. This provides the opportunity to perform an integrative data analysis, with the potential to unravel novel regulatory networks important for tamoxifen adaption.

## 2. Materials and Methods

### 2.1. Cell Cultures and Treatment

MCF-7 cells were obtained from American Type Culture Collection (ATCC, via LGC Standards, Wesel, Germany) and maintained in phenol-red-free Roswell Park Memorial Institute medium 1640 (RPMI 1640), supplemented with 10% fetal bovine serum (FBS) and stable glutamine (Biochrom, Berlin, Germany). For tamoxifen adaption, 10 nM of 4-OH-tamoxifen (Sigma-Aldrich, Taufkirchen, Germany) was added in a 1:10,000 dilution from a 100 µM stock solution in ethanol. After 12 weeks, cells were washed three times with cold phosphate buffered saline (phosphate 12.5 mM, NaCl 137 mM, KCl 2.7 mM, “PBS”) and harvested in RNA-lysis buffer (Macherey-Nagel, Düren, Germany) as part of the nucleospin miRNA preparation kit. Generation of MCF-7 Tam cells was described earlier [32,33]. Further cell lines representing major breast cancer subtypes (luminal A: T47D; HER2-overexpressing: SK-BR-3; triple negative: MDA-MB-231, Hs578T (mesenchymal-like “MSL”), MDA-MB-468 (basal-like 1 “BL1”); BRCA-methylated: UACC3199) were obtained from ATCC and grown under the same conditions as MCF-7.

### 2.2. miRNA Extraction and Analysis

miRNA was extracted using the Nuceospin miRNA kit, as described in the manufacturer’s protocol (Macherey-Nagel, Düren, Germany). Normal cDNA was synthesized as previously described using oligo dT primers, as well as random hexamer primers [32]. microRNA was determined using the Nanostring nCounter microRNA v2 human microRNA panel according to the manufacturer’s recommendations. Data were analyzed and normalized using the nSolver 4.0 software (Nanostring, Seattle, WA, USA).

For microRNA determination by qPCR, cDNA was synthesized either using the TaqMan advanced micro RNA synthesis kit (miRNA-181b-5p, -181d-5p, -92a-3p, -92a-2-5p) or a specific TaqMan microRNA assay (miRNA-1972, -455-5p, -375-3p, -181a-3p (=miR-213)) (Thermo-Fisher Scientific, Darmstadt, Germany). Real time PCR was performed in a Roche light cycler 1.0 using the LightCycler^®^ TaqMan^®^ Mastermix or LightCycler^®^ FastStart DNA Master SYBR Green I (Roche-Life Science, Mannheim, Germany). For normalization of the results, the RPL13 signal (forward primer: CCTGGAGGAGAAGAGGAAAGAGA, reverse primer: TTGAGGACCTCTTGTGTATTTGTCAA) obtained from oligo dT and random hexamer primed cDNA was used [32].

### 2.3. Gene Expression Analysis

For gene expression analysis, RT-qPCR, as well as nCounter analysis using the elements-chemistry and the 48-gene tag set for tamoxifen regulated genes, as well as the pam50 gene panel [34], was used as previously described [32].

### 2.4. Transfection with miRNA Mimics and Plasmids

For miRNA-1972, the miRidian micro RNA mimic and micro RNA mimic negative control (Horizon, Perkin-Elmer, Cambridge, UK) was used. Cells were grown to about 30% confluency in 6-well plates and the mimic (10 nM) transfected using Dharmafect2 transfection reagent, as recommended by the manufacturer (Horizon, Perkin-Elmer, Cambridge, UK). For determination of the polio-IRES/cap-translation ratio, we followed the procedure published in Vo et al.’s work in 2019 [35].

### 2.5. Western Blots

Cells were grown to medium confluency in full medium before harvesting. After washing with PBS, cells were lysed by sodium dodecyl sulfate (SDS) lysis buffer trisaminomethane (TRIS/Cl) 50 mM, pH 6.8, SDS 2%), supplied with protease and phosphates inhibitors (Sigma-Aldrich). Proteins were separated on denaturing SDS polyacrylamide gels (15%, 12% and 7.5%) and transferred to nitrocellulose membranes by semi-dry blotting [36]. After blocking in TRIS buffered saline (“TBS”: trisaminomethane 50 mM, NaCl 150 mM), supplemented with nonylphenolethoxylat-40 (NP40, 0.5%) and bovine serum albumin (BSA, 2%), the membranes were incubated with primary antibodies overnight at 4° C in a roller shaker. Then, blots were washed three times in TBS/NP40 (0.5%) BSA (0.2%) and further incubated with horseradish peroxidase conjugated secondary antibodies (Jackson-Immunoresearch, Dianova, Hamburg, Germany). After three further washes as described above, chemiluminescence was detected by enhanced chemiluminescence (ECL) reagent (Millipore, Darmstadt, Germany) in an chemostar imager (INTAS, Göttingen, Germany). Blot images were quantified using Win-Image Studio lite (Licor, Lincoln USA), version 5.2.5. Primary antibodies are listed in Table 1.

### 2.6. Proliferation Assays

For the proliferation tests, the cells were transfected at about 1/3 confluency in a 6-well plate. The next day, cells were detached, counted and 5000 cells seeded into each well of a 96 well plate. The next day, and every 24 h for a further four days, resazurin (10 µg/mL) was added and fluorescence was determined as soon as a color change became visible. Fluorescence was measured in a BMG-Labtech Clariostar reader (BMG-Labtech, Ortenberg, Germany), with excitation at 544 nm and emission set to 590 nm. All values were corrected for the results at day 1 and expressed relative to the control treatment.

### 2.7. Scratch Assays

For scratch assays, transfected cells were transferred to a 24 well plate and grown to confluency. Then, the cells were subjected to serum starvation for 24 h. A scratch was applied using a 10 µL pipette tip, the cells were subsequently washed with fresh, serum-free medium and the scratches photographed at a defined position with an inverted, phase contrast microscope using 4× and 10× objectives (Nikon TL-100), equipped with a Nikon camera system, every 24 h for 3 days. Scratches were measured at 3 positions in the microphotographs using the ImageJ software and the difference towards day 1 was calculated.

### 2.8. Target Prediction and Enrichment Analysis

Presumed miRNA targets were downloaded from Targetscan 7.2 [37] for each micro RNA, regardless of context++ score. These lists were compared to the list of 702 of the most regulated genes (at least 2-fold regulation and *p*_adj_ < 0.01) described in Porsch et al.’s work [32]. Venny 2.0 [38] was used to compare these lists and the consensus list was then submitted for enrichment analysis to the EnrichR website [39]. Here, enrichment for KEGG, reactome and GEO kinase perturbations were evaluated for miR-1972, -181-5p, -213, -375, and -455. Adjusted *p*-values (*p*_adj_ < 0.05) were assessed for determining significant enrichments. Network analysis was carried out using the Genemania website [40,41]

### 2.9. Statistics

All statistical calculations were performed using SPSS (IBM SPSS Statistics for Windows, Version 23.0. Armonk, NY, USA: IBM Corp.). For determination of statistical significance, either Student’s t-test or ANOVA with either Tamhane-T2 or LSD post-hoc analysis, depending on the presence of equal variances, was used.

## 3. Results

### 3.1. miRNAs Regulated by Tamoxifen Adaption

We first screened for 4-OH-tamoxifen-regulated microRNAs after 12 weeks of 4OH tamoxifen treatment using the nCounter microRNA panel on RNA from the three Tam-adapted cell-lines described in Porsch et al.’s work, 2019. This panel comprises about 798 miRNAs, which are detected without any further amplification techniques (Appendix A). We were able to detect the expression of 169 miRNAs. From these, the expression of five miRNAs was found to be significantly changed in the 4-OH-tamoxifen-adapted MCF-7 cell lines (Table 2). These miRNAs were analyzed further by qRT-PCR in the same cell lines to obtain further proof for the nCounter results (Table 2). This analysis was extended to selected breast cancer cell lines representing major breast cancer subtypes (Figure 1).

These cell lines were the triple negative lines MDA-MB-231 and -468, HS578T and UACC3199; the latter is also genome-hyper-methylated, and therefore BRCA defective [42]. SK-BR-3 represents an HER2/NEU over-expressing tumor and T47D and MCF-7 are ER- and PR-positive luminal A- cell lines.

The results of qRT-PCR-based determination of the miRNAs in MCF-7 and three independently obtained MCF-7-TamR cell lines correlated well with the nCounter data; however, quantitative differences were present. Compared to earlier studies, we obtained similar data as published for miR-375 [18] miR-181b [20] and miR-455 [43]. In contrast to these studies, we found additional Tam-regulated miRNAs, such as miR-1972 and miR-181d (Table 2). When comparing the microRNA expression of MCF-7 with the other cell lines, most of the microRNAs exhibited a similar pattern of expression. miR-375 showed the largest differences between the cell lines. Especially, the TNBC cell lines HS578T and MDA-MB-231 contained very low amounts of this microRNA, whereas T-47D showed a slightly higher abundance compared to the other luminal-A MCF-7 cells line. miR-455 was not detectable in the TNBC cell line MDA-MB-468.

### 3.2. Prognostic Impact of Tamoxifen-Regulated micro RNAs

Next, we investigated whether these microRNAs have an impact on breast cancer survival using the KM-plotter miRNA tool [44] using the METABRIC data [45]. For miR-1972, no reliable data were present in this dataset (Győrffy Balázs, personal communication 2020). However, using the pan-cancer dataset, a result could be obtained but should be taken with care. Indeed, all tamoxifen-regulated miRNAs had significant impact on overall survival (Table 3). A significant impact was found for all cases and persisted when we restricted the analysis to ER+ and cases that received endocrine therapy. Nevertheless, there was no correlation between up or downregulation by 4OH-tamoxifen and HR.

### 3.3. Evaluation of MicroRNA—mRNA Correlations and Enrichment Analysis

We then combined the results of the microRNA analysis with the mRNA results published earlier [32]. First, we determined the overlap between the lists of the predicted miRNA target genes and the list of 702 mRNAs that were regulated at least by a factor 2 with a *p*_adj_ < 0.01. The number of Targetscan-predicted genes ranged from 4772 (miR-1972) to 258 (miR-455). An overlap of the lists was found for 4% of the predicted targets on average (Table 4). Each of the consensus lists and a merged list were then further analyzed by enrichment analysis for KEGG, REACTOME and GEO kinase perturbations on the EnrichR website [39]. Whereas no single miRNA resulted in significantly enriched pathways, the combination of all regulated 225 target genes provided significant enrichment results. These enriched pathways included leishmania and human leukemia virus 1 infections, the Th1-, -Th2 differentiation signalling and morphine addiction (Appendix A). The GEO kinase perturbation data suggested 15 kinases, as influenced by the tamoxifen-regulated miRNAs. In the pathway enrichment analysis, these kinases were associated with major cancer pathways, such as PI3K-AKT-, ErbB-, interleukin or FoxO-signalling (Appendix A). We further investigated whether these kinases might form an interaction network using the Genemania website. Indeed, all 15 kinases could be included into a single network (Figure 2) and a further 20 proteins were added by the Genemania algorithm.

### 3.4. A miR-1972 Mimic Effects Gene Expression

We then focused further on the miR-1972, as its regulation by tamoxifen has not been described before. Targetscan analysis [37] suggested a comparatively high number of target genes and about 3.6% of these genes were indeed regulated in the tamoxifen adaption time course experiment published earlier [32]. We, therefore, decided to transfect MCF-7 as well as MCF-7-Tam cells with a miR-1972 mimic to explore the effects on gene expression further. The mimic transfection should especially counteract the tamoxifen-mediated downregulation of this miRNA in tamoxifen-adapted cells. For the analysis, we used the established nCounter technique for the tamoxifen gene-set developed earlier [32] and also the pam50 gene set, as established for the prosigna test [46,47,48]. Indeed, in both gene sets, significant gene expression changes could be identified. In case of the tam-gene set, 12 genes were significantly regulated by miR-1972 mimic transfection (Figure 3, Appendix A) in the TamR-cell line. Here, we were especially interested in genes where miR-1972 mimic transfection resulted in a more “MCF-7-like” expression. This was the case for 9 of these genes (Figure 4). MCF-7-Tam showed significant alterations in 29 genes of the pam 50 gene set, compared to the MCF-7 cell-line. Furthermore, transfection with the miR-1972 mimic resulted in 11 changes in MCF-7 and 14 in MCF-7-TamR cells, with 5 of these genes altered in both cell lines. With respect to the genes that are assigned to an intrinsic subtype [49], MCF-7-Tam cells showed changes in all subtypes (Appendix A). These were three genes in “luminal A”, nine genes in “luminal B”, three in “normal like”, seven in “Her2 enriched” and five in the “basal” subtype. The miR-1972 mimic transfection caused at least a partial reversal of the tamoxifen adaptation effect for eight of these genes (Figure 4).

### 3.5. Protein Translation Initiation and eIF-Expression in Breast Cancer Cell Lines

In addition to the pathway enrichment analysis, we were particularly interested in investigating whether tamoxifen adaption would change the mode of protein translation. Indeed, we recently observed that tamoxifen-adapted MCF-7 cell lines exhibited significant changes in the ratio of polio-IRES to cap-mediated translation. In reporter gene assays, the ratio of polio-IRES-driven firefly luciferase to cap-driven renilla-luciferase dropped in Tam-adapted cells by 57% (*p* < 0.01). We, therefore, hypothesized that this result could be mediated by differential expression of eIFs, which might be caused at least partly by the differential expression of tamoxifen-regulated micro RNAs. In our mRNA expression studies, several significant differences for the eIFs were evident, but expression differences expressed as log_2_Fc were all below ±1 (Table 5).

We then further investigated the protein abundance for the selected eIFs under the influence of 4OH-tamoxifen by Western blotting. We obtained defined Western blot signals (Figure 5 and Appendix A) for twelve eIFs (eIF2A, eIF2α, -3A, -3D, -3H, -4A1, -4B, 4E, -4G, -4H, -4EBP1, and -6) and two phosphorylated eIFs (phospho-eIF2α and phospho-eIF4E-BP1). From these, eIF2A, -3H, -4H, and -6 turned out to be significantly regulated and the ratio of phosphorylated eIF4EBP1 to eIF4BP-1 showed a statistical trend. When compared to the mRNA data (Table 5), eIF2A, -4A1 showed consistent results, whereas eIF4B showed no significant alteration in the Western blots.

In the next step, we analyzed the eIF expression in the cell lines that were investigated for micro-RNA expression (Figure 6). Here, eIF2A, eIF3H, eIF4A1, eIF4G, eIF4E, and eIF6 showed the most significant differences between the cell lines (Appendix A).

### 3.6. Correlation Analysis for eIFs and microRNAs in Breast Cancer Cell Lines

As we detected differences for 4OH-Tam-regulated miRNAs as well as for eIFs, we tested for correlations between miRNA and eIF abundance for the cell lines included into this study. The results were clustered and are shown in Figure 7. Several highly significant correlations were found, suggesting a functional or regulatory relationship between some miRNAs and eIFs. Three major clusters could be identified by this analysis. The first cluster contains peIF4EBP1 and its ratio to eIF4EBP1.

The second group contains miR-375 and -1972 together with eIF1, -2A, -4B, -4H and eIF4EBP1. The third cluster contains the miR-181 family members, as well as several eIFs. In this group, most of the highly significant correlations can be observed especially between eIF3A, eIF3H, eIF4G and the eIF2α-phosphorylation ratio.

However, when comparing these data with miRNA Targetscan predictions, only one consistent correlation was found (Table 6). miR-375 and miR-1972 are both predicted to interact with eIF4H. However, the context score for miR-1972 in particular was very low.

### 3.7. Transfections Using the miR-1972 Mimic

As we have observed several interesting correlations between Tam-regulated micro RNAs and eIFs, we intended to further support these results by transfection of MCF-7 and MCF-7-TamR cells with the miR-1972 mimic and a control RNA. Again, we determined the abundance of eIFs by Western blotting. However, here only minor changes were observed (Appendix A). Only for eIF4A1 was a downregulation found in MCF-7 (*p* < 0.1) and an upregulation found for MCF-7-Tam (*p* < 0.05). Interestingly, eIF4A1 was downregulated by tamoxifen adaption (Figure 5). eIF6 was upregulated in MCF-7-Tam cells (*p* < 0.1). The ratio of the polio-IRES to cap-mediated protein translation also remained unchanged upon transfection with the 1972 mimic (data not shown).

### 3.8. Effect of the miR1972-Mimic on Proliferation and Migration

As miR-1972 might be a factor for the aggressiveness of tumor cells, we further investigated its impact on proliferation and migration of MCF-7 and MCF-7-Tam cells. In proliferation assays, miR1972-transfected MCF-7 cells showed a significantly higher resorufin signal than control transfections, whereas the signal remained nearly unchanged in MCF-7-TamR cells (Figure 8). We also repeated these experiments in medium supplemented with 4OH-tamoxifen (10 nM); however, here MCF-7 showed only statistically insignificant effects but the negative effect on MCF-7-TamR cells became larger and statistically significant. Nevertheless, none of these differences did exceed a log_2_Fc value of ± 1.

In scratch experiments, MCF-7 cells did migrate faster than the MCF-7-TamR cells. In addition, MCF-7 cells did not show a statistically significant change in migration, whereas MCF-7-TamR cells exhibited an even reduced scratch closure (Figure 9).

## 4. Discussion

We recently reported on the changes in gene expression within 12 weeks of 4OH-tamoxifen treatment of the MCF-7 cell line, with a focus on long non coding RNAs [32]. We here analyzed gene expression in these cells further, now concentrating on micro RNAs, their relationship to the tamoxifen-regulated mRNAs and eukaryotic initiation factors. As expected from the literature, we identified tamoxifen-regulated micro RNAs, some of them already known, but especially the tamoxifen-mediated downregulation of miR-1972 represented a new finding.

The survival data obtained from the METABRIC dataset showed that all miRNAs identified as tamoxifen regulated had a prognostic impact, especially when endocrine therapy was provided (Table 3). However, data on miR-1972 might not be reliable, as the median expression was very low (Győrffy Balázs, personal communication 2020).

As the first step for an integrative analysis of mRNA and miRNA expression data, we determined whether the predicted target genes were significantly regulated by tamoxifen. The expression of about 4% of the predicted targets was indeed changed.

An enrichment analysis using these genes identified several pathways associated with infections and immune responses. In these pathways, the NF-kB-inhibitor α (NFKBIA) represented one of the major hits. This is consistent with the data suggesting that NF-kB plays an important role in acquired tamoxifen resistance [52,53,54,55,56]. The result for the pathway leading to morphine addiction seems hard to explain; however, several effects of tamoxifen on morphine responses and vice versa have been published. For example, glucuronidation of morphine and tamoxifen occurs by the same enzymes in mice, which could lead to reduced tamoxifen efficiency [57]. In mice, estrogens and tamoxifen also modified methadone responses [58]. Evidently, this cross talk should be further evaluated.

We also found several protein kinases associated with the miRNA co-regulated genes in the GEO kinase perturbation data. Most of these kinases have a well-known impact on breast cancer prognosis. These include EGFR [59], CDK8/19 [60], FGFR3 [61] and ROCK2 [62]. However, not much literature exists on breast cancer and SNRK, which should lead to further investigations. By testing for interactions using the Genemania website, all 15 kinases could be placed into a single network, which represented several cancer relevant pathways. Taken together, the results of this computational analysis are in line with earlier studies, showing that tamoxifen resistance is established by a regulatory network of micro-RNAs and signaling pathways [43,63].

As the regulation of hsa-mir-1972 seemed new to this research topic, we then focused on this miRNA by manipulating its expression by transfection with a miR-1972 mimic. We expected that this approach would provide information about whether miR-1972 is involved in the establishment of tamoxifen gene expression. Indeed, restoration of miR-1972 expression in MCF-7-Tam cells influenced the parameters that were associated with the tamoxifen adaption process. This included effects on proliferation, migration and especially changes in the expression of tamoxifen-regulated genes and genes included in the prognostically important pam50 panel.

Interestingly, miR-1972 transfection reversed the expression of a subset of tamoxifen-regulated genes into the direction of MCF-7 expression levels. We suggest that this holds for the participation of this micro RNA in the establishment of the tamoxifen-induced gene expression pattern. Furthermore, transfection with the miR-1972 mimic induced a decrease in the proliferation/vitality of tamoxifen-adapted cells, especially when 4OH-tamoxifen was present. This might be interpreted as the restoration of tamoxifen sensitivity; however, a more detailed analysis would be needed to prove this point further, especially as the effect was rather low. Migration was even further reduced in MCF-7-Tam cells, which is in line with the reduction in vitality by miR-1972 mimic transfection. In MCF-7, however, the miR-1972 mimic caused a small increase in proliferation, which was reduced in the presence of 4OH-tamoxifen. We assume that this might be caused by either the different biology of the parental cell-line or by off-target effects, caused by an unphysiologically high intracellular mimic concentration.

For miR-375, an impact on tamoxifen sensitivity has already been shown [18] and we found this micro RNA in the same eIF/miRNA cluster as miR-1972. This suggests a similar function of these two micro RNAs in breast cancer biology.

miR-1972 has rarely been investigated in breast cancer. However, this miRNA seems overexpressed in cancer tissue compared to normal tissue [64] and was, therefore, included into the dbDEMC database among the top50 breast cancer-related miRNA candidates [65]. In a comparison of 2D and 3D cell culture, however, miR-1972 was not regulated [66]. Furthermore, in an evaluation of circulating miRNAs of cancer patients and healthy donors, miR-1972 was also not conspicuous [67]. However, most interestingly, Hoppe et al. reported an upregulation of this miRNA in aromatase inhibitor (AI)-resistant MCF-7-derived cell lines [68]. This seems to be in contrast to our data but may argue for a different mechanism leading to AI resistance.

For other cancer entities, miR-1972 seems more important; especially for papillary thyroid carcinoma [69], osteosarcoma [70], ovarian cancer [71] and chronic myeloid leukemia [63]. In several of these studies, sponging of miR-1972 by non-coding RNAs has been proposed as a molecular mechanism. In osteosarcoma, differentiation antagonizing non-protein coding RNA (DANCR) decoys miR-1972 [70] but this RNA was apparently not regulated in our tamoxifen cell model. Other miR-1972-sponging linc-RNAs, such as linc00588 [72] or lnc01207 [73], were also not regulated in our cell model. Regarding gynaecological cancers, linc01125 [71] sponged miR-1972 in ovarian cancer, but again, this linc-RNA was not regulated in our tamoxifen cell model. However, such interactions might not be necessary, as miR-1972 is already down-regulated by tamoxifen itself. This might be different for dedicator of cytokinesis 9-antisense RNA2 (DOCK9-AS2) that sponges microRNA-1972 (miR-1972), leading to the upregulation of catenin β1 (CTNNB1) in thyroid cancer [69]. DOCK-AS2 was indeed moderately up-regulated in MCF-7 by 4OH tamoxifen (log_2_Fc = 0.5, p_adj_ = 0.007); however, CTNNB1 was not. Interestingly, in thyroid cancer, this DOCK2-AS2 sponging was correlated with WNT-signalling, a pathway that is also supposed to be involved in tamoxifen resistance [56,74,75]. Nevertheless, possible sponging mechanisms on miR-1972 require further analysis.

A second focus of this investigation was based upon the hypothesis that miRNA-regulated eIFs contribute to tamoxifen resistance. We have indeed found that the ratio of polio-IRES to cap-mediated translation was altered in MCF-7-TamR cells. This was accompanied by changes in the abundance of eIF2A, eIF3H, eIF4H, eiF4G and eIF6. In addition, the phosphorylation ratio of eIF2α and eIF4EBP1 was changed. This fits well with an altered start site selection in MCF-7-TamR cells. Disappointingly, miR-1972 had no effect on the polio-IRES/cap ratio. However, for eIF4A1, a modest upregulation in MCF-7 and downregulation in MCF-7-Tam-cells was observed. Additionally, eIF6 was upregulated by the miR-1972-mimic transfection in MCF-7-Tam cells. This initiation factor is not known to contribute to start site selection; however, it seems to be associated with the stress response via phosphorylation by GSK3 [76]. A miR-1972 regulation might, therefore, also reflect a stress response of the cells, which might result from the reduced proliferation/vitality caused by miR-1972 mimic transfection. Again, we propose that miR-1972 contributes to the tamoxifen effects on translation, but is not a master regulator.

## 5. Conclusions

The miRNA-1972 seems, in part, responsible for gene expression and physiological changes resulting from long-term exposure to 4OH-tamoxifen. The impact of this microRNA on the prognosis of breast cancer needs, however, further evaluation.

## Figures and Tables

**Figure 1 biomolecules-12-00916-f001:**
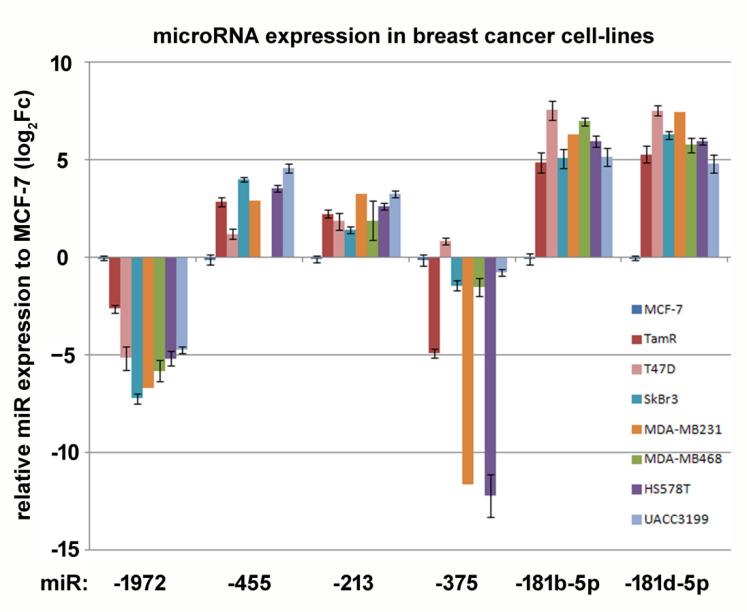
Abundance of tamoxifen-regulated miRNAs in breast cancer cell lines as determined by qRT-PCR, relative to the housekeeping gene *rpl-13*. Log_2_ of the relative expression towards MCF-7 is shown (log_2_Fc). Error bars indicate standard error (n = 3–6). Results of the ANOVA test including post-hoc analysis are presented in Appendix A.

**Figure 2 biomolecules-12-00916-f002:**
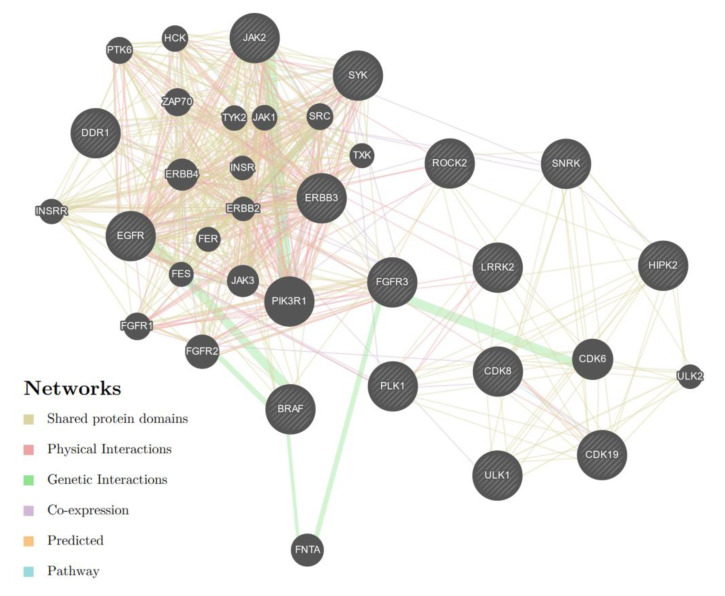
Interaction network of micro-RNA associated kinases as obtained from “Genemania” [40,41]. A total of 15 kinases associated with the tamoxifen-regulated micro RNAs (Table 4) were used for this analysis (shaded circles) and a further 20 genes were added by the Genemania algorithm.

**Figure 3 biomolecules-12-00916-f003:**
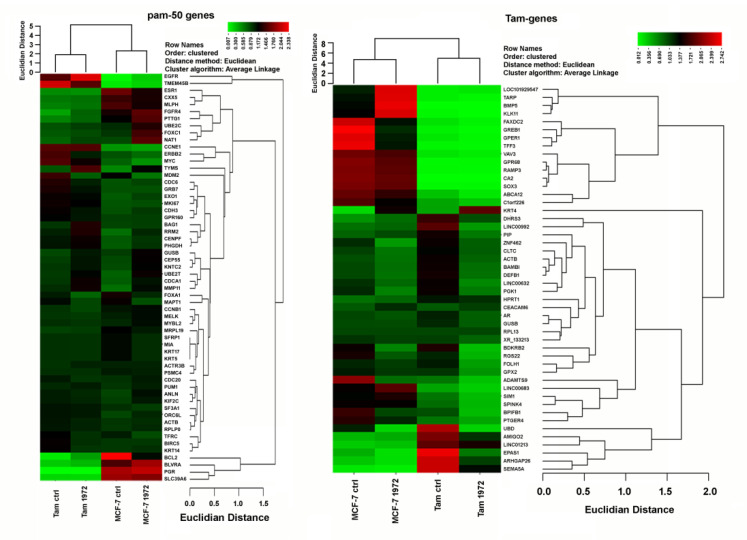
Representation of gene expression changes evoked by miR-1972 mimic transfection as heat map. Gene expression is shown relative to the average value of all experiments and shown as color-coded clustered image map (CIM) using the CIMminer web-tool [50].

**Figure 4 biomolecules-12-00916-f004:**
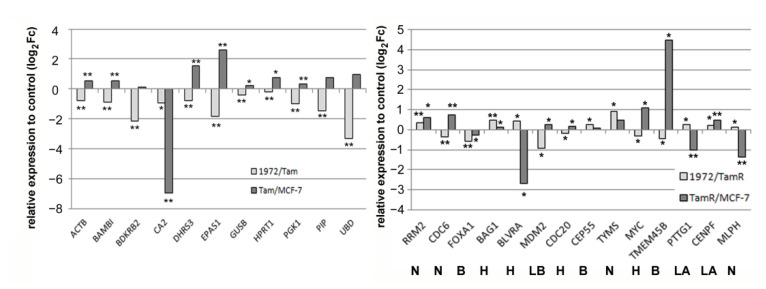
Significant changes in gene expression induced by miR-1972 mimic transfection compared to control transfections as determined by the nCounter technique. Only genes that are significantly changed in the TamR cell line are shown. Left: Tam-gene set. Right: pam50 gene set. N: normal-like, B: basal-like, H: HER2-enriched, LB: luminal B, LA: luminal A. ** *p* < 0.01, * *p* < 0.05.

**Figure 5 biomolecules-12-00916-f005:**
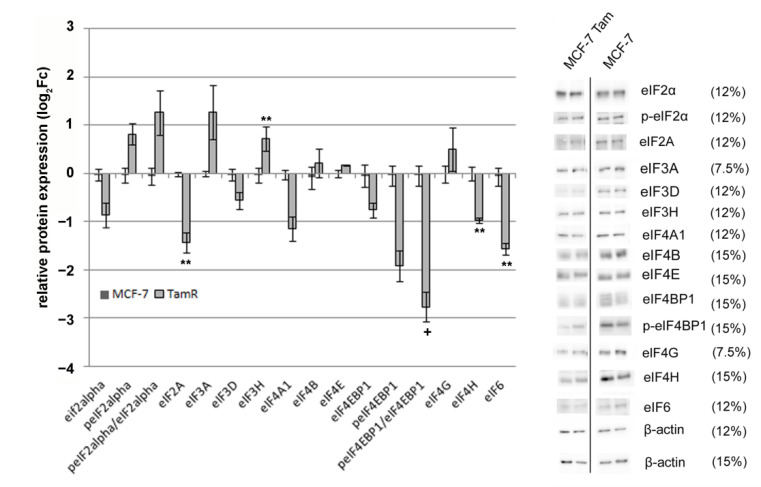
Abundance of eIF proteins in MCF-7-Tam compared to MCF-7-cells determined by Western blot. n = 3. Standard error is shown. The panel on the right shows the results of one representative experiment. The percentages refer to the density of the SDS-polyacrylamide gels used for Western blotting. Expression values (log_2_Fc), as well as statistical analysis, are shown in Appendix A. ** *p* < 0.01; + *p* < 0.1.

**Figure 6 biomolecules-12-00916-f006:**
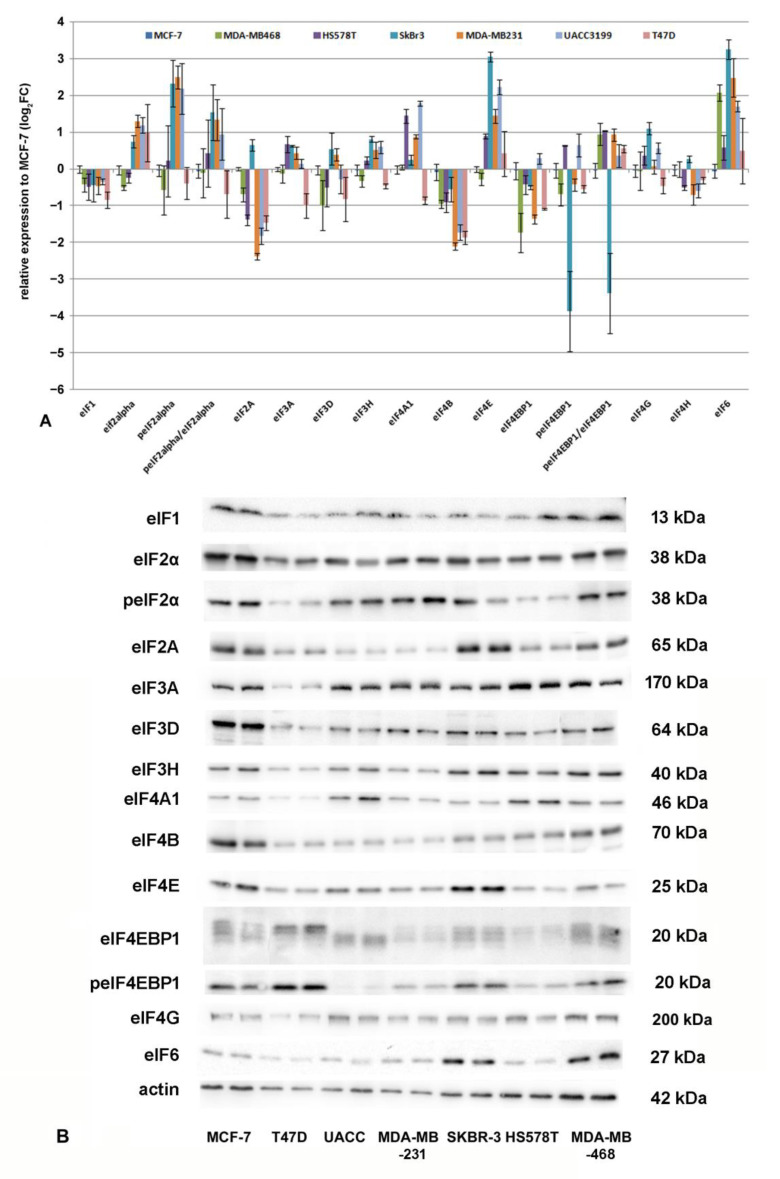
(**A**) Expression of eIFs relative to β-actin in breast cancer cell lines, as determined by Western blotting. A: Relative expression towards MCF-7 is shown as log_2_Fc. Error bars indicate standard error SEM (n = 4). (**B**) Representative Western blots of eIFs in breast cancer cell lines. Log_2_Fc values and statistical analysis by ANOVA and post hoc tests results are summarized in Appendix A.

**Figure 7 biomolecules-12-00916-f007:**
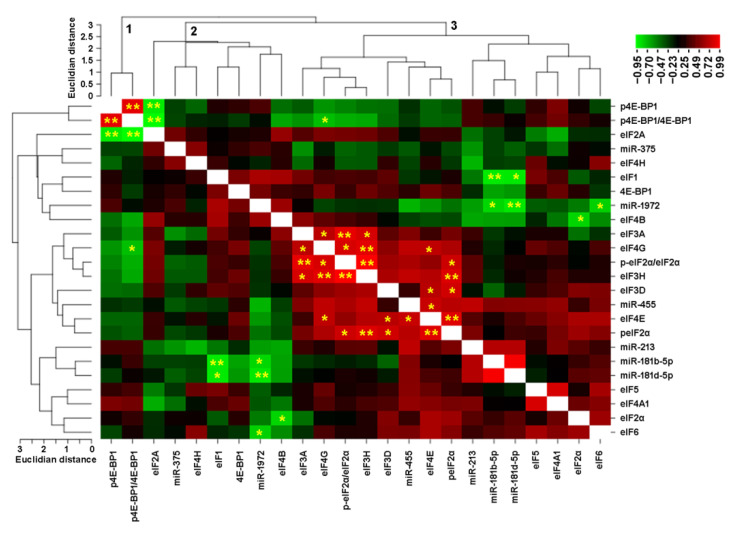
Correlation matrix of eIF-protein- and miRNA-abundance in the cell lines. A total of 8 cell lines and 12 eIFs, as well as 2 phosphorylation ratios and 6 miRNAs, were included in this analysis. eIF- and miRNA-abundance was expressed relative to MCF-7 and the Pearson correlation factor and significance determined. The correlation factors were then used for this cluster analysis using the CIMminer on-line tool [51]. Major clusters are numbered from 1 to 3. Significant correlations are indicated by *: *p* < 0.05 and **: *p* < 0.01.

**Figure 8 biomolecules-12-00916-f008:**
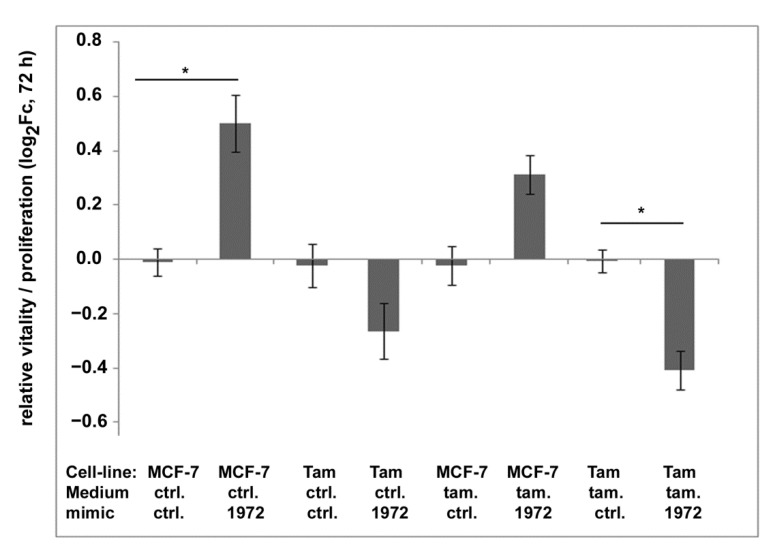
Impact of transfection with the miR-1972 mimic on viability/proliferation of MCF-7- and MCF-7-Tam cells in normal medium and medium containing 4OH-tamoxifen (10 nM). Cell viability/proliferation was determined using the resazurin assay. The fluorescence signal was normalized to the signal after seeding and shown relative to control transfections as log_2_Fc. * indicates significant differences to the control transfection determined by ANOVA and Tamhane-T2 post-hoc test (*p* < 0.05). The experiments were performed three times with 3 to 4 replicas each.

**Figure 9 biomolecules-12-00916-f009:**
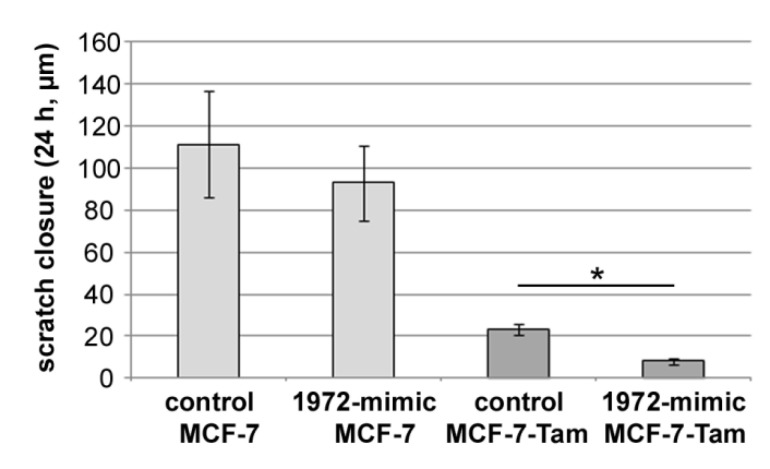
Effect of miRNA-1972 mimic transfection on migration as determined by scratch assay. MCF-7 and MCF-7-Tam cells were transfected and grown to confluence, as described in Materials and Methods. Average scratch closure after 24 h with standard error is shown. The experiment was repeated 4 times with 4 replicas each. (*: *p* < 0.05).

**Table 1 biomolecules-12-00916-t001:** Primary antibodies used in this study.

Antibody/Antigen	Supplier	Order-Nr.
eIF1	Cell Signaling	#12496
eIF2A(EPR11042)RabmAb	abcam	ab169528
eIF3A	Cell Signaling	#2538
EIF3D	GeneTex	GTX101424
eIF3H (D9C1) XP	Cell Signalling	#3413
Phospho-4E-BP1(Ser65)	Cell Signaling	#9456
4E-BP1	Cell Signaling	#9452
eIF4A1	Cell Signaling	#2490
eIF4B	GeneTex	GTX33175
eIF4E	Cell Signaling	#9742
eIF4G	Cell Signaling	#2498
eIF4H	Cell Signaling	#3469
eIF6	Cell Signalling	# 3263
β-actin	Sigma Aldrich	A5441

**Table 2 biomolecules-12-00916-t002:** nCounter data analysis showing significantly tamoxifen-regulated miRNAs and corresponding qRT-PCR data. Results of three independently generated MCF-7 TAM^R^ cell lines were combined. For nCounter analysis, three independent RNAs were used each. For qPCR, each data-point represents four independent measurements.

nCounter Results	Relative to MCF-7	
miRNA	log_2_FC	adj. *p*-value
hsa-miR-181b-5p+hsa-miR-181d-5p	5.91	0.01
hsa-miR-1972	−3.66	0.01
hsa-miR-375	−7.00	0.01
hsa-miR-181a-3p = hsa-miR-213	5.85	0.02
hsa-miR-455-5p	6.38	0.03
**qRT-PCR results**	**log_2_FC**	***p*-value**
hsa-miR-181b-5p	5.30	<0.01
miR-181d-5p	4.89	<0.01
hsa-miR-1972	−2.63	<0.01
hsa-miR-213 (181a-3p)	2.25	<0.01
hsa-miR-375-3p	−4.90	<0.01
hsa-miR-455-5p	2.87	<0.01

**Table 3 biomolecules-12-00916-t003:** Kaplan–Meier overall survival analysis for significantly 4OH-tamoxifen-regulated microRNAs using the KM-plotter website [44] and the METABRIC [45] and pan-cancer (miR-1972) dataset. Hazard ratio (HR) for high expression of the microRNA, as well as log-rank *p,* is shown. * Please note that results for miR-1972 might be unreliable, as the median expression was very low. n.a.: not available.

Micro RNA	HR (*p*) All Cases	HR (*p*) ER+	HR (*p*) Endocrine Therapy	Regulation by 4OH-Tamoxifen
miR-181a	1.5 (0.00013)	1.57 (0.00028)	1.75 (0.00028)	Up
miR-181b	1.47 (0.00029)	1.67 (3.8 × 10^−5^)	1.63 (0.00059)	Up
miR-181d	0.75 (0.0043)	0.73 (0.0065)	0.69 (0.0078)	Up
miR-375	1.29 (0.02)	1.67 (0.00043)	1.64 (0.0055)	Down
miR-455	0.78 (0.019)	0.67 (0.0019)	0.63 (0.00089)	Up
miR-1972 *	2.23 (2e-6)	n.a.	n.a.	Down

**Table 4 biomolecules-12-00916-t004:** Enrichment analysis for significant miRNA-mRNA consensus lists made by the EnrichR webtool [39]. An adjusted *p*-value < 0.05 was required for inclusion in this table.

miRNA	Targets (Targetscan)	Overlap n (%)	KEGG	REACTOME	GEO Kinase Perturbations (Up or Down)
miR-1972	4772	171 (3.6)	-	-	FGFR3 SNRK CDK19 CDK8 LRRK2 BRAF PLK1
miR-181-5p	1371	55 (4.0)	-	-	BRAF, LRRK2 JAK2 SYK DDR1 CDK19 ROCK2 ERBB3
miR-213	595	19 (3.2)	-	-	-
miR-375	304	18 (5.9)	-	-	-
miR-455	258	9 (3.5)	-	-	-
All miRNAs	6209	225 (3.6)	Leishmaniasis Morphine addiction Th1- and Th2- differentiation HTLV1 infection	-	LRRK2 FGFR3 BRAF CDK19 CDK8 ROCK2 ERBB3 ULK1 SNRK HIPK2 PLK-1 EGFR

**Table 5 biomolecules-12-00916-t005:** eIF-mRNAs exhibiting significant mRNA expression changes after 12 weeks 4OH-tamoxifen adaption [32]. Genes are sorted according to the adjusted *p*-value. Relative expression (log_2_) to untreated MCF-7 cells is shown. *, **: adjusted *p*-value < 0.5 and <0.01, respectively.

Gene	Log_2_Fc 12 Weeks
EIF4A1	−0.72 **
EIF5A	−0.73 **
EIF4EBP1	−0.72 **
EIF5A2	−0.63 **
EIF5	0.53 **
EIF2AK4	−0.41 **
EIF2AK2	−0.43 **
EIF2B3	−0.32 **
EIF1AX	−0.36 *
EIF2B2	0.45 *
EIF4B	−0.30 *
EIF4E3	0.36 *
ANKHD1-EIF4EBP3	0.30 *
EIF5AL1	0.36 *

**Table 6 biomolecules-12-00916-t006:** Significant miRNA Targetscan predictions for eIFs analyzed in this study. Targets were sorted according to the context score [37].

eIF Gene	miRNA	Total Context++ Score
EIF4E1B	hsa-miR-1972	−0.47
EIF4G3	hsa-miR-375	−0.35
EIF5AL1	hsa-miR-1972	−0.37
EIF4A2	hsa-miR-181a-5p	−0.32
EIF4A3	hsa-miR-1972	−0.32
EIF2B2	hsa-miR-181a-3p	−0.25
EIF4E	hsa-miR-1972	−0.22
EIF1	hsa-miR-1972	−0.21
EIF2S3	hsa-miR-1972	−0.25
EIF4H	hsa-miR-375	−0.18
EIF4EBP2	hsa-miR-1972	−0.17
EIF2S1	hsa-miR-181a-3p	−0.17
EIF1	hsa-miR-375	−0.15
EIF2AK2	hsa-miR-1972	−0.13
EIF2AK1	hsa-miR-181a-3p	−0.09
EIF5	hsa-miR-1972	−0.07
EIF3H	hsa-miR-375	−0.06
EIF5A	hsa-miR-1972	−0.05
EIF5B	hsa-miR-1972	−0.04
EIF4H	hsa-miR-1972	−0.02

## Data Availability

Data are contained within the article or Appendix A.

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
