# Peer review of "Relationship of micro-RNA, mRNA and eIF Expression in Tamoxifen-Adapted MCF-7 Breast Cancer Cells: Impact of miR-1972 on Gene Expression, Proliferation and Migration"

_biomolecules, 2022, doi:10.3390/biom12070916_

Round 1

Reviewer 1 Report

In this last version of the manuscript the authors improved the layout of the paper including a new title which sounds more appropriate for the results,  rearranging the results sections and, more importantly, adding a number of  bioinformatic analyses of the identified tamoxifen-regulated miRNAs (prognostic impact of microRNA, target prediction and enrichment analysis) based on available datasets.

The rest of the manuscript remains focused on the newly identified miR-1972 although no reliable data are present in the METABRIC dataset.

Effects of miR1972 mimic were analyzed after transfection in MCF-7 and tamoxifen-adapted MCF-7 cells looking at gene expression using a tamoxifen gene set developed earlier  and the pam50 gene set, in addition to the analysis of proliferation, migration and expression (mRNA and proteins) of selected eIFs. Although no striking effects were observed authors suggest a partial restoration of tamoxifen sensitivity.

Most of the observations need further evaluation as pointed out by the authors but the presented data agree with a potential role of miR-1972 in tamoxifen adaptation.

Minor points

Check if in the introduction section authors really wanted to mention miR-182-5p (ref 15) or instead mir-181-5P and if the reference is correct.

Author Response

We have decided to prepare a single point-by-point rebuttal for all remarks. We think this makes the revision more transparent.
So, there is only one file for all.

Reviewer 2 Report

In this manuscript the authors try to highlight novel mechanisms underlying the resistance to tamoxifen in breast cancer cells. The study appears overall correlative. Connections between the performed experiments are sometimes not clear. The actual relation between miR1972 and tamoxifen resistance is unclear.

Major concerns include the following:

1) The origin of the panel of breast cancer cells employed in Figure 1 should be included in Methods section.

2) GyÅ‘rffy Balázs, personal communication 2021: is there any specific reference?

3) Since the connection between miR1972 and survival could not be analyzed, the authors could look for association between miR1972 and clinic-pathological feature (if available).  

4) English editing is required.

5) What is the connection between the panel of identified microRNAs and Protein translation initiation/eIF-expression?

6) Figure 2 and Figure 3 are barely readable.

7) The actual relation between miR1972 and resistance to tamoxifen is missing. For instance, miR1972 silencing should be performed in MCF cells and sensitivity to tamoxifen should be evaluated. Also, why miR1972 forced expression is associated with increased proliferation in MCF-7 cells?

Author Response

(The authors gave the same response as above.)

Reviewer 3 Report

The manuscript "Relationship of micro-RNA, mRNA and eIF expression in tamoxifen adapted MCF-7 breast cancer cells: Impact of miR-1972 on gene expression, proliferation, and migration" is interesting to read, and the author did excellent work. However, I have some issues which need to be addressed by the author:
1. Remove period (.) from the title of the manuscript.
2. Describe cell lines, miR-1972 mimic in materials and method section

3. "About 5 % of predicted target genes were significantly altered." Incomplete information; what kind of alteration does the author want to report.
4. Several acronyms are present throughout the manuscript without their full form, i.e., HER2/NEU, FBS, PBS, and others, explain as they first appear.
5. "These are estrogen-receptor positive luminal-A and -B, Her2-overexpressing cases as well as the normal-like and basal subtype." Rephrase it.
6. "The molecular basal subtype, which overlaps to a certain extend with the immunohistochemistry based triple-negative subtype can be further divided in 6 types." Incomplete information, six types are missing.
7. In the last paragraph of the introduction, the author solely addresses the research objectives in this section. They must explain the rationale of this research.
8. Insert space in "10% and 10nM," like 10 % and 10 nM.
9. "Normal cDNA was synthesized as previously described using oligo dT as well as random hexamer primers." Either describe or cite the previously described method.

10. "For miRNA1972 the miRidian micro RNA mimic and micro RNA mimic negative control (Horizon, Perkin-Elmer) was used." Replace with, "For miRNA1972, the miRidian micro RNA mimic and micro RNA mimic negative control (Horizon, Perkin-Elmer) was used."
11. "Venny 2.0" provides a complete reference.
12. In Figure 2, the author needs to discuss 19 Kinases interactions in the main section, while the other 14 interactions will be in the supplementary section.
13. To improve the manuscript, the author can read and cite the following article "doi.org/10.1039/D1ME00183C."
14. A manuscript should be easy to read and understand, with a good flow and coherence between sentences, lacking in some parts of the manuscript. There are some minor grammatical errors, too. The author should review the manuscript to eliminate these errors (Line 46 still represents, Line 56 extent, Line 61; however, Insulin-like growth factor-1 (IGF-1), up- or down-regulation, etc.).

Author Response

(The authors gave the same response as above.)

Round 2

Reviewer 2 Report

The authors addressed the major issues of this reviewer. 

This manuscript is a resubmission of an earlier submission. The following is a list of the peer review reports and author responses from that submission.

Round 1

Reviewer 1 Report

The studies described in the manuscript aimed at investigating further tamoxifen adaptation in an MCF-7 model previously established and described by authors.

This topic has been the object of several studies describing the changes in gene expression, i.e mRNA, microRNA, lncRNA; in the present manuscript authors focus on a putative correlation between microRNA and eukaryotic initiation factors (eIFs) based on the observation that tamoxifen adaptation resulted in significant changes in the ratio of polio-IRES driven translation towards cap-dependent translation.

Analysis of microRNA expression was conducted using a Nanostring nCounter panel (798 microRNAs) and eIFs expression analyzed at both mRNA and protein levels comparing wild type MCF-7 versus 4-OH-tamoxifen adapted MCF-7 cell line (MCF-7-TamR). The miRNA screening results prompted to further investigate the expression of five miRNA in other breast cancer cell lines, of note that only two out of five microRNA were already described in earlier studies.

Correlation analysis for eIFs (proteins) and microRNAs in breast cancer cell lines outlined three cluster but miRNA Targetscan predictions indicated only one consistent prediction: miR-375 (previously described) and the newly identified miR-1972 are both predicted to interact with eIF4H.

Functional analyses were performed for miR-1972 by transfection of miRNA mimic versus control and subsequent analysis of eIFs expression (proteins), effects on viability/proliferation, migration and gene expression changes.

Although experimentally sound the results of the experiment do not allow to conclude that miR-1972 is involved in tamoxifen adaptation through regulation of eIFs nor that overexpression revert the tamoxifen-resistant phenotype.

The suggested restoration of tamoxifen sensitivity following overexpression of miR-1972 is overstated.

Author Response

PLease refer to the attasched file that contains the replay to all referees.

Reviewer 2 Report

In this manuscript, the authors investigate mechanisms of resistance to Tamoxifen in breast cancer.  The data are not clearly shown nor commented, rendering difficult to take messages home or properly judge the results. Some of the major concerns are listed below:

1) The title is not informative of the major message of the manuscript.  

2) The manuscript needs English editing.

3) Please, ameliorate the clarity of the abstract.

4) Figure 2: this figure is not clear. What is the significance of the 4 bands per row in the right panel? Please, follow the same protein order between the two panels. From those blots, this reviewer is not able to judge the presence of a correspondence between the two panels.

5) Figure 3: this reviewer has the same concerns regarding Figure 3. Correspondence between panels A and B is not clear. Also, this figure is not commented by the authors in results section.

6) Figure 4: what is the main message?

7) Why did the authors chose miR1972 for further studies? Figure 5, can the authors show the bands?

8) Figure 6 panels should be inserted in the text for proper comments. Effects of miR1972 are extremely modest.   

Author Response

(The authors gave the same response as above.)

Reviewer 3 Report

The graph abstract or summary figure will be helpful for the discussion for the micro RNA 1972 regulated by long term exposure to 4OH-tamoxifen and in responsible for the resulting gene expression- and physiological changes.

Author Response

Please refer to the attasched file that contains the replay to all referees.

Round 2

Reviewer 1 Report

Authors improved the manuscript by weakening their conclusions, showing data more clearly and adding proper comments, however the study remains descriptive with few regulatory and functional correlations.

Although these preliminary results might be of some interest for researchers in the field, further evalutions are needed, as authors point out in their response. 

Reviewer 2 Report

This reviewer believes that the results do not support the role of miRNA1972 on tamoxifen-adaption. Further investigation of this miRNA appears interesting to identify novel mechanisms of tamoxifen-adaption. However, the association with eIFs sounds extremely weak. Accordingly, transfection with miRNA mimic gave extremely modest results in relation to eIFs expression changes. The functional studies with miRNA mimic again do not support a role for this miRNA in tamoxifen-adaption (Fig 6), with modest and unexplained results (why differences in cell proliferation using medium with or without TAM?). The gene expression analysis (Figure 8 and 9) does not support a relation with eIFs. Still, the authors found interesting differences in gene expression between MCF7 and MCF7 TAM cells that could be considered for further investigation of the mechanisms mediated by miRNA 1972.